# Hepatitis C Screening and Treatment Program in Hungarian Prisons in the Era of Direct Acting Antiviral Agents

**DOI:** 10.3390/v14020308

**Published:** 2022-02-02

**Authors:** Klára Werling, Béla Hunyady, Mihály Makara, Krisztina Nemesi, Gábor Horváth, Ferenc Schneider, Judit Enyedi, Zsófia Müller, Miklós Lesch, Zoltán Péterfi, Tamás Tóth, Judit Gács, Zsuzsanna Fehér, Eszter Ujhelyi, Emese Molnár, Anna Nemes Nagy

**Affiliations:** 1Department of Surgery, Transplantation and Gastroenterology, Semmelweis University, 1082 Budapest, Hungary; 2Department of Gastroenterology, Somogy County Kaposi Mór Teaching Hospital, 7400 Kaposvár, Hungary; bhunyady@yahoo.com; 3First Department of Internal Medicine, Clinical Center, University of Pécs, 7624 Pécs, Hungary; peterfi.zoltan@pte.hu; 4National Institute of Hematology and Infectious Diseases, Szent László Site, South-Pest Central Hospital, 1097 Budapest, Hungary; michael@makara.md (M.M.); nemesikrisztina@freemail.hu (K.N.); judit.gacs@yahoo.com (J.G.); 5Hepatology Center of Buda, 1111 Budapest, Hungary; horvath.gabor@hepatologia.hu; 6Department of Infectology, Markusovszky University Teaching Hospital, 9700 Szombathely, Hungary; schfere@gmail.com (F.S.); feher.zsuzsi@gmail.com (Z.F.); 7Department of Infectology, Markhot Ferenc Teaching Hospital and Clinic, 3300 Eger, Hungary; enyedi.judit@kenessey.hu; 8Department of Infectology, Dr. Kenessey Albert Hospital, 2660 Balassagyarmat, Hungary; 9Department of Infectology, Szent György University Teaching Hospital of County Fejér, 8000 Székesfehérvár, Hungary; zsomu@freemail.hu; 10Department of Infectology, Szabolcs-Szatmár-Bereg County Hospitals Jósa András Teaching Hospital, 4412 Nyíregyháza, Hungary; leschmiklos@gmail.com; 11Department of Internal Medicine and Oncology, Semmelweis University, 1083 Budapest, Hungary; toth.tamas@med.semmelweis-univ.hu; 12Mikromikomed Ltd., 1036 Budapest, Hungary; eszter.ujhelyi@gmail.com; 13Department of Transfusiology, Semmelweis University, 1089 Budapest, Hungary; meisterwerke@gmail.com; 14Department of Health, Hungarian Prison Services, 1054 Budapest, Hungary; nemes.anna@bv.gov.hu

**Keywords:** drug users, harm reduction, hepatitis C, Hungary, inmates, prisons, screening, treatment

## Abstract

A hepatitis C virus (HCV) screening and treatment program was conducted in Hungarian prisons on a voluntary basis. After HCV-RNA testing and genotyping for anti-HCV positives, treatments with direct-acting antiviral agents were commenced by hepatologists who visited the institutions monthly. Patients were supervised by the prisons’ medical staff. Data were retrospectively collected from the Hungarian Hepatitis Treatment Registry, from the Health Registry of Prisons, and from participating hepatologists. Eighty-four percent of Hungarian prisons participated, meaning a total of 5779 individuals (28% of the inmate population) underwent screening. HCV-RNA positivity was confirmed in 317/5779 cases (5.49%); 261/317 (82.3%) started treatment. Ninety-nine percent of them admitted previous intravenous drug use. So far, 220 patients received full treatment and 41 patients are still on treatment. Based on the available end of treatment (EOT) + 24 weeks timepoint data, per protocol sustained virologic response rate was 96.8%. In conclusion, the Hungarian prison screening and treatment program, with the active participation of hepatologists and the prisons’ medical staff, is a well-functioning model. Through the Hungarian experience, we emphasize that the “test-and-treat” principle is feasible and effective at micro-eliminating HCV in prisons, where infection rate, as well as history of intravenous drug usage, are high.

## 1. Introduction

The treatment of chronic hepatitis caused by the hepatitis C virus (HCV) has changed significantly during the last few years thanks to the introduction of direct-acting antiviral agents (DAAs). The high efficacy of these therapies could potentially help to eliminate HCV infection worldwide. Currently, the main challenge for hepatologists is to identify already infected asymptomatic patients. It is estimated that currently there are around 71 million individuals left with active infection (HCV-RNA positivity); however, 50–70% of this population is unaware of their condition [1,2].

In 2016, the World Health Organization (WHO) issued a new goal: to eliminate viral hepatitis by 2030, defined as a 90% decrease in new infections and a 65% decrease in mortality [2,3,4,5]. Besides preventive and other measures, the WHO recommended the establishment of nationwide screening and therapeutic programs. Therefore, the Hungarian Minister of Human Resources founded the National Hepatitis Committee in 2018. This Committee supports the development of screening and therapeutic programs, especially in the high-risk populations that are prominent in the spread of the infection.

The HCV infection can be 10 times more frequent (16–49%) amongst inmates in prisons compared to the general population [6,7]. This difference can be explained by intravenous and intranasal recreational drug use, tattooing with unsterilized equipment and sexual promiscuity, frequent in this population. With no effective vaccine available against HCV, an obvious tool for the elimination of the virus is the prevention of spreading by the cure of infected individuals. A significant step towards this is the identification and treatment of the HCV-infected people in correctional facilities [8]. Education of the newly diagnosed patients, simplicity of the DAA treatments and their negligible side-effects all contribute to treatment completion. Successful therapy may help re-integration of inmates after completion of their sentence, without HCV-related health issues.

In Hungary, the screening and educational program aimed to eliminate HCV in correctional facilities began in 2007. By 2018, the diagnosed patients, voluntarily participating in the screening program, were treated with a combination of pegIFN and RBV. This therapeutic approach resulted in relatively low efficacy and—as a consequence of release from the facilities or re-locations—frequent early termination of therapy with the lengthy, 12-month-long procedure. Other well-known reasons for early treatment discontinuation used to be serious side effects or poor compliance of the patients. Since 2018, patients with chronic HCV infection, including inmates in correctional facilities, are treated with DAAs in Hungary.

The introduction of DAA treatments was a breakthrough in the treatment of chronic HCV infection. The main advantages of this treatment are oral administration, excellent safety profile with rare and mild side effects and outstanding efficacy, close to 100% viral clearance. The treatment strategy and treatment duration are dependent on the HCV genotype, the connective tissue remodeling in the patient’s liver and the previous treatment experience; typically, the duration of DAA therapies is 8 or 12 weeks.

Previous studies reported that DAA treatments in the inmate populations are feasible, safe, and effective [9,10,11]. Here, we report a summary of the results of our HCV screening and treatment program conducted with 5779 Hungarian inmates between 2018 and 2019.

## 2. Methods

### 2.1. Screenings

In 2018 and 2019, 5779 Hungarian inmates were screened for anti-HCV-antibody (Ab). The screening program was suspended in 2020 and 2021 due to the difficulties caused by the COVID-19 pandemic—however, treatments and follow-ups could partially be continued.

Prior to screening, hepatologists gave a briefing to inmates about hepatitis virus infections. On a voluntary basis, blood samples were taken from anti-HCV-positive patients for HCV-RNA testing and—if RNA was present—for genotyping. A participating hepatologist individually informed each HCV-RNA positive patient in the prison about the treatment, side effects and expected outcome. During the personal consultation, patients were given answers to all their questions and were able to volunteer for treatment. Prior to the initiation of the therapy, written informed consent was obtained.

Screenings were performed by the medical staff at the prisons. At present, 21 hepatologists in 26 prisons in Hungary (84% of Hungarian prisons) manage and administer treatments with the help of healthcare staff of the facilities.

As a screening method, anti-HCV was determined from venous blood by Adaltis Eigan HCVab (v. 4) ELISA assay kit (Adaltis S r.l., Milano, Italy, Ref No: 071067). Upon anti-HCV-Ab positivity, HCV-RNA PCR test was performed using the DNA Technology PCR system and the Cepheid HCV Viral load test (Cepheid Company GmBH Sunnyvale, CA, USA, Ref No: CE-GXHCV-VL-CE-10). Genotyping was also performed before initiation of treatment, using the DNA Technology kit (DNA Technology LLC. Moscow, Russia, Ref No: ABHCG1, ABHCG2). Tests were performed by the Laboratory of the South-Pest Hospital and the Mikromikomed Ltd. (Mikromikomed Ltd.—Budapest, Hungary) —in an accredited molecular virology center in Hungary.

### 2.2. Treatments

There are 33 centers in Hungary, which cover the whole country for HCV treatments. Inmates were treated by a hepatologist from the Hepatology Centre in the area, who visited the institute once a month and was available for consultation by phone at any time. The medical staff of the correctional facility administered the medication and monitored the patients.

Prior to initiating treatment, laboratory tests were performed to assess blood counts, liver enzyme, serum albumin and bilirubin levels, INR and renal function parameters, in addition to any potentially relevant differential diagnostic tests according to the opinion of the participating hepatologist. These tests were performed by laboratories assigned to the prison. (Table 1).

We used the FIB4 score to determine the degree of connective tissue remodeling with cut-off values <1.45 (F0-1), 1.45–3.25 (F2/3), >3.25 (F4/6). Classical Child–Pugh score (based on serum bilirubin and albumin levels, INR, encephalopathy and ascites) was used to evaluate liver compensation status.

The Hepatitis Therapeutical Committee (appointed by professional organizations) reviewed all treatment applications and advised on reimbursement for the National Health Insurance Fund [12].

Applied treatment strategies: grazoprevir–elbasvir (n = 197), glecaprevir–pibrentasvir (n = 4), sofosbuvir–ledipasvir (n = 13), sofosbuvir–velpatasvir (n = 25), ombitasvir/paritaprevir/ritonavir + dasabuvir (n = 22) with or without ribavirin according to national guidelines [13]. According to the national guideline, duration of the therapy was typically 8 or 12 weeks, exceptionally 16 or 24 weeks. Sustained viral response (SVR) was confirmed by a below-the-limit detection level of HCV-RNA 12 or 24 weeks after the end of treatment (EOT). Patients have reimbursed access to medication through the National Health Insurance Fund of Hungary.

### 2.3. Data Collection

Primary patient and therapy-related data were retrospectively collected from the national treatment database (HepReg) [12]. The Health Department of the Hungarian Prison Service and the Health Service of the prisons also participated in the data collection. Additional data were provided by the treating physicians from the health records on file in the respective Hepatology Centers.

### 2.4. Ethics Committee Approval

Approval of the Scientific and Research Ethics Committee of Hungarian Medical Research Council was obtained (approval number: IV/9969-1/2021/EKU, date of approval: 16 December 2021) to confirm that the study meets national and international research ethics guidelines.

## 3. Results

### 3.1. Screening Test Results

Disposal of screened individuals is shown in Figure 1. During the screening period of 2018–2019, 20,538 inmates were detained, and 5779 (28.1%) inmates participated voluntarily in the HCV screening program. Out of them, HCV-RNA positivity was found in 317/5779 patients (5.5%).

### 3.2. Patient Characteristics

Patients’ characteristics are shown in Table 1. Outcomes of previous interferon-based treatments are shown in Table 2. Of the 261 treated patients, FIB4 score was <1.45 in 232 (88.9%) patients, between 1.45 and 3.25 in 25 patients (9.6%) and >3.25 (suggesting liver cirrhosis) in four patients (1.5%). No patient with hepatic decompensation was identified based on the Child–Pugh parameters. The median (mean) virus titer was 160,583 (1,286,636) IU/mL. Five patients had HBV co-infection, but no HIV co-infected was identified.

The HCV genotype distribution was as follows: G1 with unknown subtype: 10 (3.8%), G1a: 128 (49.0%), G1b: 93 (35.6%), G1c: 1 (0.4%), G3: 28 (10.7%), G4: 1 (0.4%).

Ninety-nine percent of HCV-infected inmates admitted to a history of intravenous drug use.

### 3.3. Treatment Outcome

DAA treatment was initiated in 261/317 (82.3%) HCV-RNA-positive patients. The treatment was not commenced in 56 (17.7%) patients for various reasons shown in Table 3. Out of the 261 initiated therapies, it was concluded for 220 (84.3%) patients to date, while 41 (15.7%) treatments were ongoing at data lock.

However, EOT+24 HCV-RNA results are available only for 126 patients. Out of these 126 patients, 122 presented a negative HCV-RNA result, and four patients were positive, resulting in a per-protocol SVR rate of 96.8% from the available data. The EOT+24 result is unavailable for 135 (51.7%) patients. The main reasons for missing data are patients’ transport into a non-participating correction facility release of inmates or restrictions related to the COVID-19 pandemic (Figure 2).

### 3.4. Safety

No serious adverse event was reported by either the treating hepatologists or the health services of the prisons. Reported mild-to-moderate side-effects, occurring in more than two patients, were minor fatigue and headache. No other drug-related symptoms or changes in laboratory parameters were detected.

## 4. Discussion

Screenings and DAA treatments performed in the Hungarian prisons validate the high effectiveness, success rate and safety of this approach. Inmates are afraid of stigmata, exclusion, and the side-effects of the treatment. Lack of trust and confidence further aggravates this problem. Patient education and personal contact with the physician who is to treat them later significantly increased patient compliance. The acceptance of the DAA therapy was significantly better compared to the previously used pegIFN+RBV combination therapy (82% versus 63%, respectively; unpublished data). Twenty-four percent of the patients participating in this study discontinued a previous pegIFN+RBV therapy. Patients favor DAA treatment due to the oral administration, the short treatment duration and the minor side-effects. Multiple patients presented in this dataset knew previously about their HCV infection, however, they refused earlier available pegIFN+RBV therapy.

DAA treatment was not initiated for 17.7% of the diagnosed patients in this study. The main reasons behind this were other illnesses that contraindicated the treatment, patients’ transport into other correctional facilities, completion of the prison sentence or that the COVID-19 pandemic abrupted the treatment. Only 1.26% of the diagnosed patients refused the therapy. Hence, prison is a suitable microenvironment for the treatment of HCV, especially with DAA therapies where the short treatment duration allows the full regimen to be completed while inmates are serving their sentence. Negligible side effects also improve patients’ compliance. Furthermore, patients are treated in prisons under controlled conditions and the effectiveness of the drugs allows successful viral clearance. Thus, they cannot transmit the disease later on, when released [14,15,16].

Despite the high HCV prevalence in the prison population, the screening and treatment program is not in practice in several countries. According to a recent survey that covered 25 European countries, HCV treatment programs in prisons are operated in only 21 of them [17]. The treatment of these patients is an important tool to achieve the HCV elimination goal of WHO by 2030. Infected inmates can transmit the disease to other inmates as well as to the general population after their release, through intravenous- or intranasal drug use, tattooing with unsterilized equipment or by sexual promiscuity. Almost 99% of the inmates presented in this study admitted previous intravenous drug use, and more than 90% of them were tattooed.

The outcomes are negatively affected by the release or transfer of the inmates before finishing the therapy or having been evaluated for SVR. An article published by Aspinnal et al. reported that 74% SVR was achieved in Scottish prisons using DAA treatments when patients spent the entire period of the treatment in the same prison, versus 45% in those who were released and 59% in those who were transferred to another prison [18]. Studies show that only 9% of former inmates continue their treatment after release [19]. This is consistent with our observations: for most of those who were transferred to another prison or released, treatment was not continued and the chance of HVC clearance was reduced.

The effectiveness of the “test-and-treat” method was evaluated in a Spanish prison between May 2016 and July 2017. Treatment of patients was directed by teleconsultation from a hospital. The anti-HCV test was performed in nearly 100% of inmates, and 10.2% were confirmed to be HCV-RNA positive. Sixty-nine of 86 patients started treatment, and an SVR of 96.9% was achieved with DAA treatment–comparable to our per-protocol SVR. By the end of treatment, there were no HCV-RNA positive inmates in the prison and no re-infection, or de novo infection was observed until the end of the observation period. These data also highlight the effectiveness of the “test-and-treat” strategy in prisons [20].

Pontali and colleagues reported a study about the treatment of 142 patients from 25 different prisons in Italy [21]. Similarly to our results, 94.4% of the patients finished the DAA treatment. The most frequent HCV genotypes reported were G1a and G3. They reported a 90.8% SVR, which is lower compared to our results. The possible reasons behind this might be that 76.1% of the enrolled patients had hepatic cirrhosis, a significantly higher proportion than in our study population. They reported that the connective tissue remodeling was higher than F2 in 9.58% of the patients, and out of this, 1.53% of the patients had a score higher than F4.

A prospective study conducted in Portugal between 2017 and 2020 reported that despite 22.4% of the patients having stage F3-4 hepatic fibrosis, the SVR was 99% [22]. The high SVR rate of the treatment in patients with advanced liver fibrosis shows that young patients without any other chronic diseases in prison, under controlled conditions, have good compliance and an excellent viral clearance rate can be achieved.

Unfortunately, the EOT+24 weeks HCV-RNA results are unavailable for 51.7% of our patients, mostly due to release, transportation to other detention facilities or the difficulties caused by the COVID-19 pandemic. Similar results are shown by Bhandari et al. in North East England [23]. They reported that 41% of the patients were lost in the follow-up period. In this study, 71% of the PCR-positive patients accepted the therapy and the available data presented an SVR rate of 87%.

The literature data and our results show that intravenous drug use and tattooing are the most important causes of infection [24,25]. The proportion of PWIDs and/or tattooed individuals was about 90–99% in our patient population. Participants usually refused to answer questions about their sexual habits. Previous literature data also suggested that drug use among inmates was very high, with some data suggesting that 47.7% of anti-HCV-positive patients in prison were PWIDs. A Canadian study found that HCV prevalence in inmates was significantly higher in PWID patients than in non-drug users (51.0% versus 2.4%). In this study, the proportion of tattooed individuals was 37.2% [26].

Due to the high proportion of drug users, needle and syringe programs (NSPs) and opioid substitution therapy (OST) play major roles in controlling the spread of the infection. There is an OST program but no NSP program in Hungarian prisons. The situation is similar in most European countries. According to a survey, only 8% of countries have NSPs and 44% have OST in prisons [17].

Previous studies reported genotype distribution of 5.6% G1a, 84.6% G1b and 1.8% G3 in the Hungarian population (without prison inmates) [27]. The prevalence of genotypes is different in our prison cohort: 49.0% G1a, 35.6% G1b, and 10.7% G3, this difference reflects that the infection is connected to intravenous drug use among inmates. According to a survey, genotypes G1a and G3 are the most common ones amongst Hungarian drug users [28].

Multiple modeling studies reported that curative treatment of HCV in the prison population could lead to a decrease in prevalence in the general population [29,30,31,32]. These studies underline the importance of screening and DAA therapy programs in the prison populations.

In this relatively young population (median age 34 years) with typically no or mild fibrosis (<F2 in 88.9% of patients), consequently with no or exceptional history of hepatocellular carcinoma, with a large proportion being vaccinated against hepatitis B (immunization data not collected in this study, but mandatory and completed for individuals born after 1986 in Hungary), long term liver outcomes might be presumably excellent in this population, and long term follow-up, recommended for patients with advanced fibrosis, cirrhosis, previous HCC, and/or no viral clearance by several international organizations might be omitted or cut short [33]. Depending on the volume of the screened and treated inmate population, medical consequences (in particular, liver cirrhosis, hepatic decompensation, hepatocellular carcinoma, liver transplantation, reduced life-expectancy) and expenses related to their HCV infection might be reduced significantly with such efforts in the long term, in addition to epidemiology benefits.

## 5. Limitations

The data processed in our article were compiled from multiple sources including records of colleagues, the HepReg database, and the databases of the prisons, and are often incomplete due to the nature of real-life prison conditions. Patients were in many cases transferred to another facility during treatment where no treatment continuations and/or follow-ups were provided; others were released. Therefore, blood collections were often missed, and treatments were not completed; also, in many cases, blood collection was not performed at EOT+12 or EOT+24 weeks, thus SVR could not be confirmed.

We have no data on the percentage of patients who were re-infected after their release. If a patient was convicted again, they were most often admitted to another institution where there was no screening program. Due to the lack of access to the national health register in prisons, these patients were not identified.

Another shortcoming of our article is that HCV screening was only occasionally complemented with HBV and HIV screening. This was partly due to financial reasons, but also owing to the fact that in Hungary vaccination against HBV at the age of 13 is mandatory since 1998. Since inmates are mainly young people, they are likely to have been vaccinated.

Finally, a major limitation was the pause of our program in 2020 and 2021 due to the COVID-19 pandemic.

## 6. Conclusions

In Hungary, the HCV screening and treatment program in prisons, now running for over 15 years, is a well-functioning system with active participation from hepatologists all over the country. Patients are treated locally in the prison, which enables close cooperation between doctor and patient and ensures optimal treatment results. No such long-running, large-scale prison program has been reported from other countries. Despite recommendations of professional guidelines, no or only partial HCV screening and treatment programs are being executed in many countries. Previous studies have shown that the most effective practical approach is to screen and treat in prisons. Through the results demonstrated in the Hungarian Prison Service program, we are able to demonstrate that the “test-and-treat” policy is feasible and effective at micro-eliminating HCV from a high-risk group, prison inmates, where the proportion of prior intravenous drug users consequently the rate of HCV infection are excessively high.

Shorter duration, fewer side effects, and the effectiveness of the new DAA treatments improved patients’ compliance. The time spent in prison was, in most cases, sufficient to clear the HCV infection. After release, however, most patients failed to contact the treating physician to continue therapy or to show up for the follow-up. This confirms the paradoxical fact that in this particular high-risk group, therapy can be more efficiently delivered while the patients are incarcerated since as inmates they can be more readily persuaded, are being kept under controlled conditions, and have more interest in cooperating. In the prisons, treatments were administered in close cooperation between the prison medical staff and appointed hepatologists.

Our goal in the future is to draw the attention of decision-makers to the necessity of governmental support for the program. Additionally, we are advocating an amendment of regulations in force to mandate the screening of inmates for HCV, HBV, and HIV once they start their sentence.

## Figures and Tables

**Figure 1 viruses-14-00308-f001:**
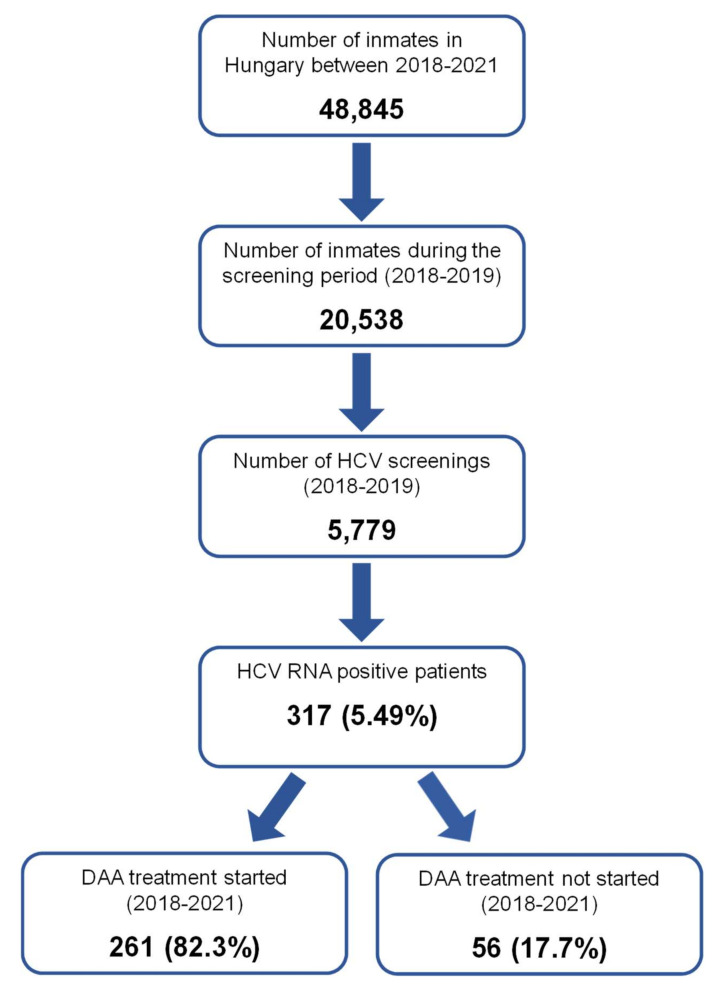
Screening and treatment in the Hungarian Prison Service between 2018 and 2021.

**Figure 2 viruses-14-00308-f002:**
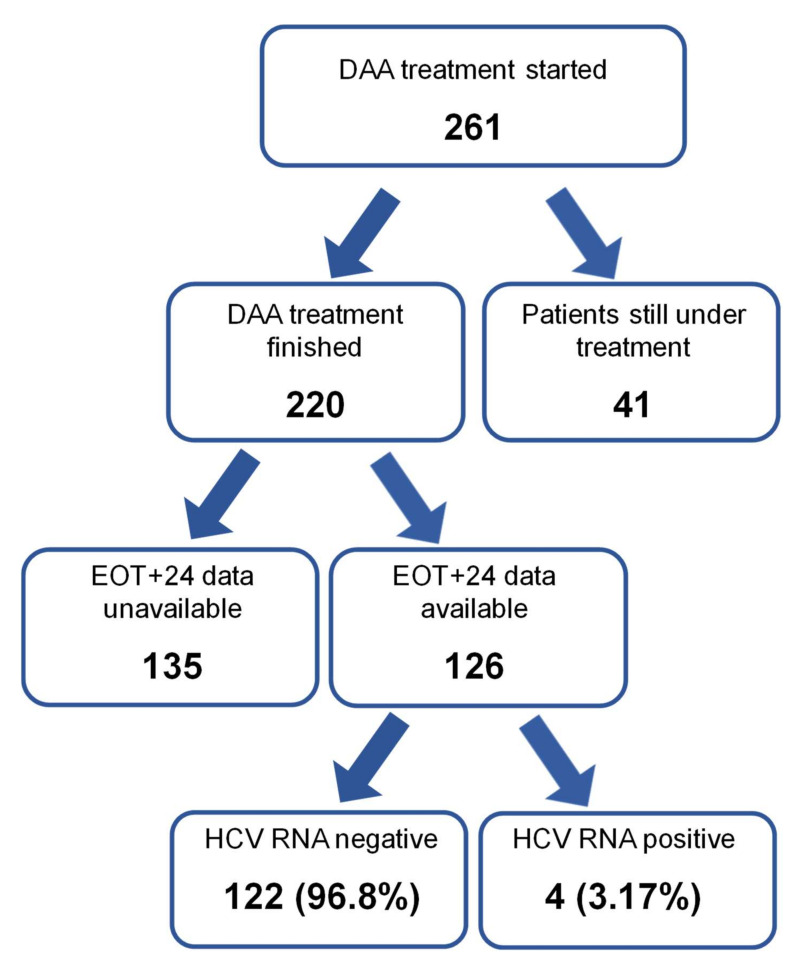
Outcome of the DAA treatment.

**Table 1 viruses-14-00308-t001:** Data of treated patients.

Parameter	Data
Number of Patients Treated with DAA	261
Male/female, n/n	220/41
Age, median/mean (range), year, (number of available data)	34/34.3 (17–64), (n = 261)
History of intravenous drug use, n (%), (number of available data)	252 (99%), (n = 255)
ALT, median/mean (range), IU/L, (number of available data)	74/94.0 (10–430), (n = 261)
AST, median/mean (range), IU/L, (number of available data)	41/50.9 (12–189), (n = 261)
Serum bilirubin, median/mean, µmol/L, (number of available data)	10/11.1 (2.2–29), (n = 261) *
Serum albumin, median/mean, g/L, (number of available data)	47/46.7 (25.0–57.6), (n = 261) *
INR, median/mean, (number of available data)	1.1/1.0 (0.8–1.5), (n = 261)
Hemoglobin, median/mean, g/L, (number of available data)	155/153.6 (68–197), (n = 256)
Platelet count, median/mean, G/L, (number of available data)	231/232.4 (73–497), (n = 256)
Absolute neutrophil cell count median/mean, G/L, (number of available data)	4.2/4.5 (1.1–9.5), (n = 256)
Estimated glomerular filtration rate, median/mean, mL/min/1.73 m^2^, (number of available data)	90/85.0 (6–137), (n = 201) *
Virus titer, median/mean (range), IU/L, (number of available data)	160,583/1,286,636 (17–90,000,000), (n = 261)
Genotype distribution	
G1 with unknown subtype, n (%)	10 (3.8%)
G1a, n (%)	128 (49.0%)
G1b, n (%)	93 (35.6%)
G1c, n (%)	1 (0.4%)
G3, n (%)	28 (10.7%)
G4, n (%)	1 (0.4%)
FIB4 score median/mean (range), (number of available data)	0.72/1.23 (0.24–35.28), (n = 261)
FIB4 < 1.45, n (%)	232 (88.9%)
FIB4 1.45–3.25, n (%)	25 (9.6%)
FIB4 > 3.25, n (%)	4 (1.5%)
HBV coinfection, n, (number of available data)	5 (n = 258)
HIV coinfection, n, (number of available data)	0 (n = 245)

* Patients with Gilbert’s syndrome, nephrotic syndrome, renal protein loss, on hemodialysis and/or with other comorbidities were eligible to participate upon discretion of treating physician and review of Therapeutic Committee. ALT: alanine aminotransferase, AST: aspartate aminotransferase, DAA: direct-acting antiviral agents, G: genotype, PCR: polymerase chain reaction.

**Table 2 viruses-14-00308-t002:** Outcomes of previous interferon-based treatments in study cohort.

Status	Number of Patients
Naïve, n (%)	219 (84.5%)
Partial responder, n (%)	3 (1.1%)
Null responder, n (%)	6 (2.3%)
Relapsed/re-infected, n (%)	8 (3.1%)
Previous treatment suspended, n (%)	25 (9.6%)

Naïve: previously not received any antiviral therapy; relapsed/re-infected: previously responded to interferon-based therapy then relapsed or re-infected; partial responder: previous interferon-based therapy caused a decrease in viral load, but patient never became PCR negative; null responder: HCV-RNA positivity on the 24th week after the initiation of interferon-based treatment.

**Table 3 viruses-14-00308-t003:** Reasons for not commencing treatment.

Reason	Number, n
Hepatocellular Carcinoma, n	1
Multiple hepatic metastases, n	1
Decompensation of non-hepatic disease, n	1
Passed away before treatment, n	2
COVID-19 pandemic-related, n	1
Refused treatment, n	4
Released or moved to another facility, n	30

## Data Availability

Anonymous patient data presented in this study are available on request from the corresponding author. The data are not publicly available due to GDPR reasons.

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
