# Peer review of "Hepatitis C Screening and Treatment Program in Hungarian Prisons in the Era of Direct Acting Antiviral Agents"

_viruses, 2022, doi:10.3390/v14020308_

Round 1

Reviewer 1 Report

In the present mannuscript, Werling and colleagues reported the results of a multicenter hepatitic C screening and treatment program in Hungarian prisons conducetd between 2018 and 2021. The study included 26 prisons in Hungary, with 5779 out of 20538 inmates voluntarily screened for anti-HCV-antibodies. Among anti-HCV+ subjects, 317 (5.49%) resulted HCV RNA+. Of these subjects, 261 (82.3%) underwent DAA treatment; per protocol analysis showed that 96.8% achievded SVR (EOT+24 HCV RNA -negative). This study showed the effectiveness of a screening and treatment program among subjects at high risk of HCV infection; these results are relevant, expecially considering WHO aim of eliminate viral hepatitis by 2030. The study design is appropriate and the manuscript is well written.

Below some minor comments.

1) Table 1. Due to the not-normal distribution of continuous variables (i.e. age, ALT, HCV RNA, and FIB-4), I suggest to report such variables as median and range (or IQR). Furthemore,  I suggest to report next to each considered variable, the number of patients in which the data were available. Authors may consider to add the number in brackets. For instance: Median ALT, IU/mL (range) [...].

2) Reference section. According to JCM style, please report for each refeence the first 10 authors followed by et al. 

Reviewer 2 Report

Wering et al. reported that HCV screening and treatment program in Hungarian inmate in the era of direct acting antiviral agents. This seems an important article in this area. Authors may achieve 90% decrease in new infections. Authors should give some comments to achieve decrease in mortality, especially due to HCC.

See: Kanda T, et al. APASL HCV guidelines of virus-eradicated patients by DAA on how to monitor HCC occurrence and HBV reactivation. Hepatol Int. 2019 Nov;13(6):649-661. doi: 10.1007/s12072-019-09988-7. Epub 2019 Sep 20. PMID: 31541423

Author Response

“Authors should give some comments to achieve decrease in mortality, especially due to HCC.”

In this relative young population (median age 34 years) with typically no or mild fibrosis (<F2 in 88.9% of patients), consequently with no or exceptional history of hepatocellular carcinoma, with a large proportion being vaccinated against hepatitis B (immunization data not collected in this study, but mandatory and completed for individuals born after 1986 in Hungary), long term liver outcomes might be presumably excellent in this population, and long term follow up, recommended for patients with advanced fibrosis, cirrhosis, previous HCC, and/or no viral clearance by several international organizations might be omitted or cut short.[33] Depending on the volume of screened and treated inmate population, medical consequences (in particular, liver cirrhosis, hepatic decompensation, hepatocellular carcinoma, liver transplantation, reduced life-expectancy) and expenses related to their HCV infection might be reduced significantly with such efforts on long term, in addition to epidemiology benefits.
33. Kanda, T.; Lau, G. K. K.; Wei, L.; Moriyama, M.; Yu, M-l.; Chuang, W-L.; Ibrahim, A.; Lesmana, C. R. A.; Sollano, J.; Kumar, M.; et al. APASL HCV guidelines of virus-eradicated patients by DAA on how to monitor HCC occurrence and HBV reactivation. Hepatol Int., 2019, 13, 649-661. doi: 10.1007/s12072-019-09988-7.

Reviewer 3 Report

This is a very interesting study by Werling et al, providing data from a past HCV screening program performed within Hungarian prisons. The data is well presented and the study design is sound. 

No major negative points can be raised. Incomplete data and several methodological shortcomings are acknowledged by the authors.

One minor point of improvement might be inserting a synthetic table with biological data at admission (blood counts, liver enzyme, serum albumin and bilirubin levels, INR and renal function parameters) as the authors state that these values were collected prior to treatment initiation.

Author Response

“One minor point of improvement might be inserting a synthetic table with biological data at admission (blood counts, liver enzyme, serum albumin and bilirubin levels, INR and renal function parameters) as the authors state that these values were collected prior to treatment initiation.”

See revised table 1